# A Study of Different Types of Air Pollutants on the Efficiency of China’s Hotel Industry

**DOI:** 10.3390/ijerph16224319

**Published:** 2019-11-06

**Authors:** Xiaoying Guo, Wei Wei, Yang Li, Lei-Ya Wang

**Affiliations:** 1Department of International Economics and Trade, School of Economics, Tianjin University of Commerce, Tianjin 300134, China; gxy@tjcu.edu.cn; 2Department of Economics, School of Economics, Tianjin University of Commerce, Tianjin 300134, China; weiw2016@tjcu.edu.cn; 3New Huadu Business School, Minjiang University, Fuzhou 350108, China; 4Institute of Business and Management, National University of Kaohsiung, Kaohsiung 811, Taiwan; m1053601@nuk.edu.tw

**Keywords:** Data envelopment analysis (DEA), visible air pollutant, invisible air pollutant, China’s hotel industry, bootstrapped truncated regression

## Abstract

This study focuses on visible and invisible air pollutants and their impacts on China’s hotel industry. Overall, visible air pollutants may block the sights and sceneries and worsen the quality of visitors’ sensory experiences, and invisible air pollutants are unlikely to result in the same perceptions and sensations. Hence, different types of air pollutants may have various impacts on the hotel industry’s operational performance. We employed a bootstrapped truncated regression model to investigate whether different types of air pollutants had distinctive impacts on the hotel industry. The dataset consisted of 31 provinces of China for the period 2012–2015. Empirical results indicate that visible air pollutants significantly decrease the operational efficiency of China’s hotel industry, while invisible air pollutants insignificantly affect the hotel industry.

## 1. Introduction

Air quality can influence the process of making travel decisions as well as shape the competitiveness of tourist destinations [1,2]. Unlike residents in polluted areas, tourists can easily substitute other destinations for their vacation. Touring to an air-polluted destination would give travelers unpleasant experiences; thus, they would not likely come back to and/or recommend that destination [3,4]. Air pollution can be generally divided into two types: visible and invisible. Visible air pollutants, such as dust, obviously impacts the quality of tourism by blocking the view of sceneries and worsening the sensory experience, while invisible air pollutants, such as nitrogen oxides, are unlikely to create such perceptions and sensations. According to our review, there is no literature distinguishing the impacts between visible and invisible air pollutants on tourism. Hence, the objective of this study is to investigate whether different types of air pollutants have distinctive impacts on the performance of the hotel industry, thus helping to fill the gap in understanding the effects of environment quality on tourism.

China is noted for its beautiful scenery, abundant heritage resources, and distinctive cultures. Since 1990, China has been the third largest inbound tourism destination globally [3], even becoming the second largest tourism market in 2018 [5]. Since the launch of its widely known economic reforms in 1978, China’s tourism industry has experienced rapid growth [6]. The “golden week holiday”, first set up in 1999, further accelerated the development of China’s travel and tourism industry [7]. In addition, local governments have turned literary places into tourism resources, proving the value of literary tourism with respect to culture and tourism [8,9]. However, with rapid urbanization and industrialization, China’s air pollution has gradually turned more serve and prominent. Many studies have indeed presented the negative impacts of air pollution on tourism [1,10,11,12]. For instance, haze-related air pollution has notably decreased the number of tourists visiting Brunei Darussalam [10], and the problem of impaired visibility has significantly affected visitation numbers to the Great Smoky Mountain National Park in the U.S. [11]. Through questionnaire surveys from potential tourists, Zhang et al. [1] discovered that haze pollution in Beijing has spurred many potential tourists to cancel travel plans and/or to reschedule their travel time to avoid terrible weather.

Poor air quality not only negatively alters people’s quality of life and causes health problems, but it may also form heavy smog that obstructs the view of tourism locations. Since travelers generally visit tourist destinations only for a few days, air pollution may not influence their health in such a very short period. Hence, air pollutants affect the hotel industry mainly through visitors’ perceptions and sensations on tourist destinations. However, visible air pollutants, such as smoke and dust, obviously impact the quality of tourism by impeding the view of sceneries and deteriorating sensory inputs, while invisible air pollutants, such as sulfur dioxide and nitrogen oxides, are unlikely to generate such perceptions and sensations. Hence, different types of air pollutants may have distinctive impacts on the operational performance of the hotel industry. This implies that, when evaluating the influence of air pollution, we should carefully distinguish the impacts of different types of air pollutants on the hotel industry.

Data envelopment analysis (DEA), proposed by Charnes et al. [13], is essentially a linear programming model to evaluate the efficiencies of decision-making units (DMUs) by calculating the best multiplier for inputs and outputs. Since it can deal with multiple inputs and outputs without assuming any particular functional frontier form, it has been widely applied in many different fields, including the tourist hotel industry [14,15,16,17,18]. Many studies have analyzed how environmental variables influence operating efficiencies by the two-stage approach, employing DEA to obtain efficiency scores in the first stage and then regressing efficiency scores on environmental variables in the second stage. Most of them have specified the tobit model in the second stage by observing that several efficiencies are equal to unity, suggesting a probability mass at a value of one and a concept of latent variables [19,20,21,22,23,24,25,26,27]. Simar and Wilson [28] argued that it is primarily an artifact to decide whether efficiency is equal to one and not the property of latent variables, and so the second stage should utilize a truncated specification; in addition, the dependent variable, estimated by the DEA model, is serially correlated, and the random distance in the second stage is also correlated with environmental variables. Therefore, Simar and Wilson [28] introduced bootstrap procedures to overcome the above problems. Our study also uses the bootstrapped truncated regression model, as proposed by them, to investigate whether different types of air pollutants have distinctive impacts on the hotel industry.

## 2. Hypotheses and Research Framework

Environmental quality not only essentially shapes the competitiveness of tourist destinations, but it also crucially influences the process of making travel decisions [29,30,31]. Travelling to an air-polluted destination would give tourists unpleasant experiences; thus, they would unlikely revisit and/or recommend that destination. Unlike residents in the polluted areas, tourists are basically “voting with their feet” and easily can substitute other tourist destinations for their vacation. Several studies also found the negative impacts of air pollution on tourism. For instance, haze-related air pollution has notably decreased the number of tourists visiting Brunei Darussalam [10], while the problem of impaired visibility significantly has affected visitation numbers to the Great Smoky Mountain National Park [11].

Air pollution can form a heavy smog that obstructs the view of tourism locations. as Air pollution can negatively affect people’s quality of life and results in health problems. Sightseers usually stay in tourist destinations only for a couple of days; thus, their health may not be affected significantly by air pollution in such a very short time. Hence, air pollutants influence the hotel industry primarily through visitors’ sensations and perceptions on tourist destinations. Visible air pollutants, such as dust, can impede sights and sceneries and, thus, worsen tourists’ sensory experiences; while invisible air pollutants, such as sulfur dioxide, are unlikely to generate such perceptions and sensations. Hence, visible air pollutants should have a larger impact on the hotel industry versus invisible air pollutants. We thereby propose Hypothesis 1 as follows.

**Hypothesis** **1.**
*The impact of visible air pollutants is larger than that of invisible pollutants on the hotel industry.*


The tourist hotel sector presently benefits from rapid growth and prosperity of the overall tourism industry, but in the face of greater competition resulting from more providers and potential entrants, managers of tourist hotels are continuously seeking out appropriate strategies to enhance profitability and growth. Rumelt [32] suggested that when facing keen competition, decline of sales, a relatively mature market, and other threats, a firm should adopt a diversification strategy to overcome said problems. Since diversification helps transfer strategic assets and core competencies to other business units to create synergy and, thus, effectively improves operational performance [33,34,35], it has become an important strategy for a firm to increase profit margins and market shares [36]. In adopting a diversification strategy, tourist hotels not only can increase revenue, but they can also pursue stability in operating performance [37,38,39].

The revenue generated by guest rooms is the core source of profit for the hotel industry. In recent years, the proportion of revenue obtained from the food and beverage (FB) industry has been on an upward trend [40], especially in Asian regions where the importance of FB revenue is no less than that of guest rooms [41]. Therefore, the hotel industry now commonly implements revenue diversification strategies. Hence, revenue diversification should contribute more to chain hotels than to independent hotels. We now propose Hypothesis 2 as follows.

**Hypothesis** **2.**
*Revenue diversification benefits the performance of the hotel industry.*


Based on previous studies, we added transport infrastructure, location, and year dummies as control variables [16,42,43,44] in order to truly reveal the effects of air pollutants and revenue diversification on the performance of the hotel industry. Figure 1 illustrates the conceptual framework of this study.

## 3. Methodology

DEA was initially proposed by Charnes et al. [13], which was called the CCR model based on the concept of technical efficiency by Farrel [45]. Many studies have used a two-stage approach to analyze how environmental variables influence operating efficiencies, which is achieved by employing DEA to obtain efficiency scores in the first stage and then regressing the efficiency scores on environmental variables in the second stage. Most studies have specified the tobit model in the second stage by observing that several efficiencies are equal to unity [19,21,23,25,27]. Simar and Wilson [28] argued that it is primarily an artifact of the finite samples of the DEA model to decide whether efficiency equals one and not the property of latent variables. Hence, the appropriate approach in the second stage should be a truncated regression model. In addition, the dependent variable, estimated by the DEA model, is serially correlated, and the random distance in the second stage is also correlated with environmental variables. Simar and Wilson [28] introduced the bootstrapped truncated regression model to overcome the above problems.

Assume that there are *N* decision-making units (DMUs). Each DMU employs *P* inputs, denoted by $\underset{\sim }{x}=[x_{1},\cdots ,x_{P}{]}^{\prime }\in {ℜ}_{+}^{P}$, to produce *M* outputs, denoted by $\underset{\sim }{y}=[y_{1},\cdots ,y_{M}{]}^{\prime }\in {ℜ}_{+}^{M}$. The reciprocal of the output-oriented CCR model for DMU *j* can be written as:(1)$$\underset{\theta_{j},\lambda_{1},\lambda_{2},\cdots ,\lambda_{N}}{\max}\;\begin{array}{cc} & \theta_{j} \end{array},$$
subject to
$$\sum_{n=1}^{N}\lambda_{n}\;x_{pn}\leq x_{pj},\begin{array}{cc} & \;\;\;p=1,\;2,\;\cdots ,P \end{array},\;\sum_{n=1}^{N}\lambda_{n}\;y_{mn}\geq \;\theta_{j}\;y_{m\;j},\begin{array}{cc} & m=1,\;2,\;\cdots ,M \end{array},\;\lambda_{1},\lambda_{2},\ldots ,\lambda_{N}\geq 0;\;\;\begin{array}{cc} & \theta_{j}\;\mathrm{is}\;\mathrm{free} \end{array}.$$

The optimal value of $\theta_{j}$, denoted by ${\hat{\theta }}_{j}$, is the efficiency measure of DMU *j*. It can be shown that ${\hat{\theta }}_{j}$ is between 0 and 1. When ${\hat{\theta }}_{j}$ = 1, it indicates that the production point is on the frontier and, hence, is a technically efficient DMU; otherwise, its production point is inside the frontier and, thus, a technically inefficient DMU.

The CCR model assumes that production exhibits constant returns to scale (CRS), which is only appropriate when all DMUs are operating at an optimal scale. Banker et al. [46] extended the CCR model to account for variable returns to scale (VRS), called the BCC model. Mathematically, the BCC model is modified easily from the CCR model by adding the convexity constraint $\sum_{n=1}^{N}\lambda_{n}=1$. Note that this convexity constraint essentially guarantees that an inefficient DMU is only benchmarked against DMUs with similar sizes.

Many studies have computed the estimates of $\hat{\theta }$ by the DEA model in the first stage and then regressed them on *q* environmental variables $\underset{\sim }{z}=[z_{1},\ldots ,z_{q}{]}^{\prime }\in {ℜ}^{q}$ in the second stage:(2)$${\hat{\theta }}_{n}={\underset{\sim }{\beta }}^{\prime }{\underset{\sim }{z}}_{n}+\varepsilon_{n}\geq 1\begin{array}{cccc} , & & & n=1,2\ldots ,N \end{array},$$
where $\underset{\sim }{\beta }$ is a *q* × 1 vector of parameters, and $\varepsilon_{n}$ is a continuous iid random variable with mean zero and constant variance $\sigma_{\varepsilon }^{2}$. Studies have viewed ${\hat{\theta }}_{n}$ as the realization of latent variables and estimated Equation (2) by the tobit regression method. Simar and Wilson [28] indicated that there are several problems with this empirical model. First, the tobit specification is motivated by the observation that several efficiencies are equal to unity, suggesting a probability mass at one and a concept of latent variables. However, deciding whether efficiency equals one is primarily an artifact of the finite samples of the DEA model and not the property of latent variables. Hence, the second stage should utilize the truncated specification, and $\varepsilon_{n}$ is distributed as $N(0,\;\sigma_{\varepsilon }^{2})$ with left-truncation at $\left(1-{\underset{\sim }{\beta }}^{\prime }{\underset{\sim }{z}}_{n}\right)$.

Second, the dependent variable ${\hat{\theta }}_{n}$ cannot be observed directly and has to be estimated by the DEA model in the first stage. Hence, not only is ${\hat{\theta }}_{n}$ serially correlated, but the random disturbance $\varepsilon_{n}$ in Equation (2) is also correlated with environmental variables ${\underset{\sim }{z}}_{n}$. Finally, the DEA estimator obtained from the DEA model under mild assumptions is consistent, but it converges slowly at the rate $N^{\;-2/\left(P+M+1\right)}$, known as the curse of dimensionality [47], suggesting that even though the maximum likelihood (ML) estimators of $\underset{\sim }{\beta }$ in the second-stage regression are consistent, they are unlikely to obtain reliable conference intervals. Since the structures of the above phenomena are not known to be associated with an extremely slowly convergent rate, Simar and Wilson [28] proposed a bootstrap procedure to overcome these problems.

The efficiency score ${\hat{\theta }}_{n}$, by construction, is biased downward [48]. Although it is consistent, the bias will disappear at the slow rate of $N^{\;-2/\left(P+M+1\right)}$. Simar and Wilson [28] suggested another bootstrap procedure to correct this bias. The bias-corrected estimator of ${\hat{\theta }}_{n}$ is:(3)$${\hat{\hat{\theta }}}_{n}={\hat{\theta }}_{n}-\bar{BIAS}({\hat{\theta }}_{n}),$$
where $BIAS\;({\hat{\theta }}_{n})\equiv E\;({\hat{\theta }}_{n})-\theta_{n}$, and $\bar{BIAS}({\hat{\theta }}_{n})$ is the bootstrap bias estimate.

We described the double bootstrap procedure, proposed by Simar and Wilson [28], in the following steps.

{1}Use all DMUs to calculate ${\hat{\theta }}_{n}$ (*n* = 1, 2,…, *N*) by the DEA model.{2}Acquire the ML estimates $\left(\hat{\underset{\sim }{\beta }},\;{\hat{\sigma }}_{\varepsilon }\right)$ of $\left(\underset{\sim }{\beta },\;\sigma_{\varepsilon }\right)$ in the truncated regression ${\hat{\theta }}_{n}={\underset{\sim }{\beta }}^{\prime }{\underset{\sim }{z}}_{n}+\varepsilon_{n}$, where $\varepsilon_{n}$ are iid $N(0,\;\sigma_{\varepsilon }^{2})$ with left-truncation at $\left(1-{\underset{\sim }{\beta }}^{\prime }{\underset{\sim }{z}}_{n}\right)$ by all inefficient DMUs, and ${\hat{\theta }}_{n}>1$, *n* = 1, 2,…, *J* (< *N*).{3}For each DMU (*n* = 1,…, *N*), loop the following four steps ({3.1}–{3.4}) ***B***_1_ times to obtain a set of bootstrap estimates ${\left\{{\hat{\theta }}_{nb}^{*}\right\}}_{b=1}^{B_{1}}$:
{3.1}Draw $\varepsilon_{n}^{*}$ randomly from $N(0,\;{\hat{\sigma }}_{\varepsilon }^{2})$ with left-truncation at $\left(1-\hat{{\underset{\sim }{\beta }}^{\prime }}{\underset{\sim }{z}}_{n}\right)$;{3.2}Compute $\theta_{n}^{*}=\hat{{\underset{\sim }{\beta }}^{\prime }}{\underset{\sim }{z}}_{n}+\varepsilon_{n}^{*}$;{3.3}Set ${\underset{\sim }{x}}_{n}^{*}={\underset{\sim }{x}}_{n}$ and $\;{\underset{\sim }{y}}_{n}^{*}={\hat{\theta }}_{n}{\underset{\sim }{y}}_{n}/\theta_{n}^{*}$;{3.4}Calculate ${\hat{\theta }}_{n}^{*}({\underset{\sim }{x}}_{n},\;{\underset{\sim }{y}}_{n})$ by the DEA model with technology $\left(X^{*},\;Y^{*}\right)$, where $X^{*}=[{\underset{\sim }{x}}_{1}^{*},\ldots ,{\underset{\sim }{x}}_{N}^{*}]$ and $\;Y^{*}=[{\underset{\sim }{y}}_{1}^{*},\ldots ,{\underset{\sim }{y}}_{N}^{*}]$.{4}For each DMU (*n* = 1,…, *N*), compute the bias-corrected estimate ${\hat{\hat{\theta }}}_{n}=\;{\hat{\theta }}_{n}-\bar{BIAS}({\hat{\theta }}_{n})$, where $\bar{BIAS}({\hat{\theta }}_{n})=B_{1}^{-1}\sum_{b=1}^{B_{1}}{\hat{\theta }}_{nb}^{*}-{\hat{\theta }}_{n}$.{5}Get the ML estimates $\left(\hat{\hat{\underset{\sim }{\beta }}},\;{\hat{\hat{\sigma }}}_{\varepsilon }\right)$ of $\left(\underset{\sim }{\beta },\;\sigma_{\varepsilon }\right)$ in the truncated regression of ${\hat{\hat{\theta }}}_{n}$ on ${\underset{\sim }{z}}_{n}$ with left-truncation at $\left(1-{\underset{\sim }{\beta }}^{\prime }{\underset{\sim }{z}}_{n}\right)$.{6}Loop the following three steps ({6.1}–{6.3}) ***B***_2_ times to obtain a set of bootstrap estimates ${\left\{{\left({\hat{\hat{\underset{\sim }{\beta }}}}^{*},\;{\hat{\hat{\sigma }}}_{\varepsilon }^{*}\right)}_{b}\right\}}_{b=1}^{B_{2}}$:
{6.1}For each DMU (*n* = 1,…, *N*), draw $\varepsilon_{n}^{**}$ randomly from $N(0,\;{\hat{\hat{\sigma }}}_{\varepsilon }^{2})$ with left-truncation at $\left(1-\hat{\hat{{\underset{\sim }{\beta }}^{\prime }}}{\underset{\sim }{z}}_{n}\right)$;{6.2}For each DMU (*n* = 1,…, *N*), compute $\theta_{n}^{**}=\hat{{\hat{\underset{\sim }{\beta }}}^{\prime }}{\underset{\sim }{z}}_{n}+\varepsilon_{n}^{**}$;{6.3}Find the ML estimates $\left({\hat{\hat{\underset{\sim }{\beta }}}}^{*},\;{\hat{\hat{\sigma }}}_{\varepsilon }^{*}\right)$ of $\left(\underset{\sim }{\beta },\;\sigma_{\varepsilon }\right)$ in the truncated regression of $\theta_{n}^{**}$ on ${\underset{\sim }{z}}_{n}$ with left-truncation at $\left(1-{\underset{\sim }{\beta }}^{\prime }{\underset{\sim }{z}}_{n}\right)$.{7}Use the original estimates $\left(\hat{\hat{\underset{\sim }{\beta }}},\;{\hat{\hat{\sigma }}}_{\varepsilon }\right)$ and the bootstrap values obtained from step {6} to construct estimated confidence intervals for each element of $\underset{\sim }{\beta }$ and for $\sigma_{\varepsilon }$.

## 4. Empirical Analysis

### 4.1. Data and Input–Output Variables

The dataset, obtained from The Yearbook of China Tourism Statistics published by the National Tourism Administration of the People’s Republic of China during 2012–2015 [49,50,51,52], consisted of 31 provinces in the country. All nominal variables were deflated by the GDP deflator with 2011 as the base year. We considered three inputs—number of employees, number of guest rooms, and fixed assets—and three outputs—revenues of guest rooms, revenues of food and beverage (FB), and other revenues. Table 1 reports the descriptive statistics of the inputs and outputs used in the analysis.

The input and output variables in the DEA model should satisfy the property of isotonicity—that is, increased inputs cannot reduce outputs. Table 2 presents the Pearson correlation coefficients of the input and output variables. All values were significantly positive at the 0.1% level of significance, indicating that our selected input and output variables indeed met the property of isotonicity.

### 4.2. Empirical Specifications and Results

Balk [53] argued that actual technology should be treated as VRS, and that even though a DMU is technically efficient under VRS, one can additionally increase its productivity by improving the operating scale along the VRS frontier. Hence, this study employed an output-oriented BCC model to calculate efficiencies in the first stage.

Table 3 summarizes the original DEA and bias-corrected efficiency scores. The average efficiency without bias correction, obtained from Equation (1), was 0.7891 ($\hat{\theta }$ = 1.2673), with 0.8308, 0.7604, and 0.7541 being the average efficiency of the east, central, and west regions, respectively. In addition, 26 DMUs were efficient ($\hat{\theta }$ = 1) in operating on the production frontier: 15 were in the east region, 7 in the central region, and 4 in the west region.

The efficiency $\left(1/\hat{\theta }\right)$ obtained from Equation (1), however, was biased upward [48]. Although this efficiency score is consistent, the bias will disappear at a slow rate. Following the suggestion of Simar and Wilson [26], this study used 200 replications for the first bootstrap of the double-bootstrapped procedure to obtain the reciprocal of bias-corrected efficiency $\hat{\hat{\theta }}$.

Table 3 indicates that the overall average efficiency was 0.7208 ($\hat{\hat{\theta }}$ = 1.3874), where the average was 0.7688 for the east region, 0.699 for the central region, and 0.6414 for the west region. Note that since the original DEA efficiency scores were biased upward [48], they were always larger than the bias-corrected efficiency scores. Furthermore, the maximum value of the bias-corrected efficiency scores was 0.9615 ($\hat{\hat{\theta }}$ = 1.04), indicating that there was no efficient DMU in our sample after being bias-corrected.

The second stage employed the bootstrapped truncated regression model, proposed by Simar and Wilson [28], to investigate how different types of air pollutants affected operating efficiencies in China’s hotel industry. The dependent variable was the reciprocal of bias-corrected efficiency $\hat{\hat{\theta }}$, obtained from the first bootstrap of the double-bootstrapped procedure. According to data availability from The Yearbook of China Tourism Statistics, we included two invisible air pollutants (sulfur dioxide (SO_2_) and nitrogen oxides (NO_X_)) and one visible air pollutant (covering smoke and dust) in the regression model.

Some studies have created a revenue diversification index from the revenues of guest rooms and revenues of FB [40,54]. In addition to these two kinds of revenues, tourist hotels can earn other revenue, including, but not limited to, the lease of store spaces, laundry services, swimming pools, ball courts, barbershops, beauty salons, and bookstores [55]. We, thus, constructed the revenue diversification index by revenues of guest rooms, revenues of FB, and other revenues. Let *R_i_* be the ratio of the *i*th revenue to total revenue. The entropy index of revenue diversification is expressed as:(4)$$\begin{array}{cc} ERev= & -\sum_{i=1}^{3}R_{i}\ln R_{i} \end{array}.$$

The larger the value of *ERev* is, the higher the degree of revenue diversification.

Previous studies have suggested that explanatory variables, such as transport infrastructures, location dummies, year dummies, and, affect performances in the hotel industry [16,42,43,44,56,57,58]. We added these covariates into the empirical model as control variables in order to truly reveal the influence of different types of air pollutants on the operational efficiency of China’s hotel industry. We included two variables to measure transport infrastructures: road intensity, measured by the ratio of the total distance of roads (km) to the area (10,000 km^2^) of provinces, and railroad intensity, measured by the ratio of the total distance of railroads (km) to the area (10,000 km^2^) of provinces. Since both roads and railroads offer comfortable and convenient transportation for tourists, we expected they would contribute positively to the efficiency of China’s tourism industry.

We added regional dummies (*DE* and *DC*) and year dummies (*Year*13, *Year*14, and *Year*15) to control heterogeneities among DMUs. The dummy variable *Di* = 1 if the province is in region *i*, and 0 otherwise. We treated the west region as the reference region, where *i* = *E* for the east region and *i* = *C* for the central region. Taking 2012 as the reference year, we added three year dummies: *Year*13 = 1 if the year is 2013, and 0 otherwise; *Year*14 = 1 if the year is 2014, and 0 otherwise; and *Year*15 = 1 if the year is 2015, and 0 otherwise. Hence, the empirical model is specified as:(5)$$\begin{array}{l} {\hat{\hat{\theta }}}_{n}=\beta_{0}+\beta_{1}SO_{2}{}_{n}+\beta_{2}NO_{x}{}_{n}+\beta_{3}S\&D_{n}+\beta_{4}ERev_{n}+\beta_{5}RoadI_{n}+\beta_{6}RailI_{n}+\beta_{7}DE_{n}\\ +\beta_{8}DC_{n}+\beta_{9}Year13_{n}+\beta_{10}Year14_{n}+\beta_{11}Year15_{n}+\varepsilon_{n},\begin{array}{cc} & n=1,2,\ldots ,N, \end{array} \end{array}$$
where $\varepsilon_{n}$ is distributed as $N(0,\;\sigma_{\varepsilon }^{2})$ with left-truncation at ${\hat{\hat{\theta }}}_{n}\geq 1$ for each *n*. Table 4 presents the definition and sample means of the variables used in the bootstrapped truncated regression model.

Referring to the situation where there is either an exact or an approximately exact linear relationship among explanatory variables, multicollinearity is an undesirable situation, since it misleadingly inflates the standard errors. Thus, it makes some variables statistically insignificant when they otherwise should be significant. The variance inflation factor (VIF), based on the coefficient of determination (*R*^2^) of auxiliary regressions, is generally used to detect multicollinearity. Chatterjee and Price [59] suggested that values in excess of 10 are problematic. The last column of Table 4 shows that all values of VIF were less than 9.1. Hence, we concluded that all explanatory variables used in our empirical model did not exhibit the problem of multicollinearity.

Table 5 presents the bootstrapped truncated regression results. Taking advice from Simar and Wilson [28], we used 2000 replications in the bootstrapped procedure to construct estimates of confidence intervals. Columns 2–8 present the estimated results for the bootstrapped truncated regression model, among which column 5 shows a mean estimate obtained from the bootstrap set with 2000 replications; columns 2 and 8 are, respectively, the lower and upper boundaries of the 99% confidence interval; columns 3 and 7 are, respectively, the lower and upper boundaries of the 95% confidence interval; and columns 4 and 6 are, respectively, the lower and upper boundaries of the 90% confidence interval. Note that the covariate in the empirical model can improve technical efficiency (or reduce ${\hat{\hat{\theta }}}_{n}$) if its coefficient is negative.

The empirical results of the bootstrapped truncated regression show that the estimated coefficients of *S*&*D*, *ERev*, *RoadI*, *TrainI*, *DC*, *Year13*, *Year14*, and *Year15* were significantly different from 0, among which *S*&*D*, *Year*14, and *Year*15 were at the 1% level of significance, and the others were at the 5% level of significance. The estimated coefficient of *S*&*D* was significantly positive at the 1% level, while both *SO*_2_ and *NO*_x_ were insignificantly different from zero at the 10% level, indicating that different types of air pollutants may have distinct impacts on the hotel industry. We may conclude that visible air pollutants significantly decrease the efficiency of China’ hotel industry, but invisible air pollutants do not significantly influence the hotel industry of China, thus supporting hypothesis 1. However, the coefficient of revenue diversification (*ERev*) was insignificantly positive at the 10% level; thus, Hypothesis 2 was not supported.

## 5. Discussion

The impact of environment quality on tourism has long been emphasized by tourism researchers [29,30]. Travelers are increasingly seeking a tourist destination with a high-quality environment [60], as air-polluted destinations result in a diminished quality of their travel experiences. China is a large country with beautiful scenery, plentiful heritage resources, and distinctive cultures, but inbound tourist arrivals into China have been declining in recent years [12]. Li [61] suggested that increasing levels of air pollution are undermining its popularity as a travel destination. The literature also found the negative impact of poor air quality on economic development as a result of decreased tourism activity [10,62]. As a continued decline in international travelers to China is possible if no efforts are taken to deal with the issue of its air pollution, it is becoming more vital for the tourism sector to take actions against air pollution. Hence, some researchers have suggested that tourism can play an important role as a promoter for clean air and as a leading industry for applying clean technology that decreases damaging emissions [12].

It is generally true that adopting appropriate activities can help fight air pollution, but various types of air pollutants may have distinct impacts on the hotel industry; for instance, visible air pollutants may block the sights and sceneries and, thus, worsen tourists’ sensory experiences, while invisible air pollutants are unlikely to allow the creation of such perceptions and sensations. Previous research studies have not investigated whether different types of air pollutants have distinctive impacts on tourism. Our empirical results present that visible air pollutants (smoke and dust) significantly decrease the efficiency of China’s hotel industry, whereas invisible air pollutants (sulfur dioxide and nitrogen oxides) insignificantly affect it. Thus, when evaluating the influence of air pollution, we should carefully distinguish among the impacts of different types of air pollutants on the hotel industry. In other words, to efficiently battle air pollution, the first priority for the tourism sector is to effectively reduce the emission levels of visible air pollutants and dust, since decreasing emission levels of invisible air pollutants such as sulfur dioxide and nitrogen oxides may not contribute positively to the efficiency of the hotel industry.

Benefitting from the expansion of global tourist markets, the tourist hotel industry has also experienced rapid growth. In the face of swift development and increasing intra-industry competition, diversification has become an important strategy of the tourist hotel industry to reduce risk, increase revenue, and improve performance [37,39]. Previous studies have also found a positive relationship between revenue diversification and performances in the tourist hotel industry [38,39,40,63]. However, Table 5 indicates that revenue diversification cannot effectively contribute to the performance of China’s hotel industry, thus not supporting Hypothesis 2. This may result from the specific characteristics of Chinese culture. Chinese food is famous all over the world and consists of a surprising range and variety of ingredients. Different regions have distinct styles of cooking; in addition, ingredients are based on the natural and agricultural products of each region. There are many famous historic restaurants in China, and for many tourists, one of the necessary itineraries when visiting a scenic area is to taste specific local food. However, restaurants in a hotel generally need considerable capital to operate and have to compete with those famous historic ones. Hence, revenue diversification may not contribute positively to the operating efficiency of China’s hotels. This may suggest that China’s hotel industry should concentrate on the main revenue generated by guest rooms and focus on how to increase and/or stabilize this core business.

The estimated coefficients of all transport infrastructures, regional dummies, and year dummies are significantly differently from zero at the 5% level, except *DE*. Hence, other things being equal, we claim that (1) transport infrastructures contribute positively to the operating efficiency of China’s hotel industry; (2) the performance of China’s hotel industry in the western region is worse than that in the central region, but insignificantly different from that in the east region; and (3) China’s hotel industry in 2012 significantly outperformed the other study time periods.

## 6. Conclusions

Air-polluted destinations lead to diminished quality of tourists’ experiences and, thus, decrease the operational efficiency of local hotels. This study has, thus, employed a bootstrapped truncated regression model to analyze whether different types of air pollutants have distinct influences on China’s hotel industry. We found that visible air pollutants (smoke and dust) significantly decreased the operational efficiency of the country’s hotel industry, whereas invisible air pollutants (sulfur dioxide (SO_2_) and nitrogen oxides (NO_X_)) insignificantly affected the hotel industry. In addition, transport infrastructures can effectively promote the operating efficiency of China’s hotel industry, while revenue diversification insignificantly affects the operational performance of hotels. 

This study found that visible air pollutants significantly impacted the efficiency of China’s hotel industry, whereas invisible air pollutants insignificantly altered it. The same methodology can also be employed in other areas to investigate whether different types of air pollutants may have distinctive impacts on the operational performance of the hotel industry. In addition, future studies can also employ conditional nonparametric frontier approaches to investigate how external environmental variables, which are neither inputs nor outputs, influence the operational performance of DMUs.

## Figures and Tables

**Figure 1 ijerph-16-04319-f001:**
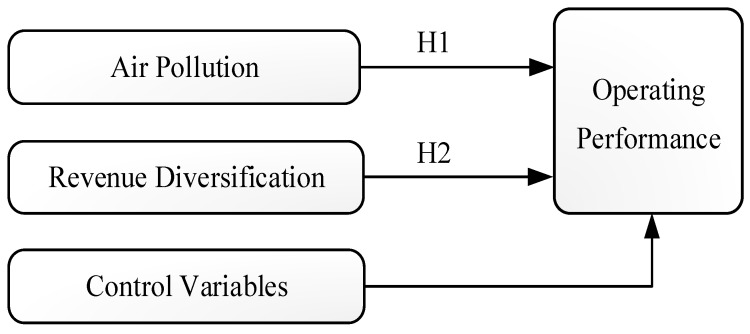
The conceptual framework.

**Table 1 ijerph-16-04319-t001:** Statistics of Inputs and Outputs (2011–2013).

	Mean	Std. Dev.	Min	Max
**Input Variables**				
Number of employees	46,764.52	33,963.99	3594.00	154,157.00
Fixed assets (RMB million)	13.59	12.73	1.42	59.42
Number of guest rooms	47,352.77	30,358.70	6413.00	146,820.00
**Output Variables**				
Revenue of guest rooms (RMB million)	261.51	250.42	21.67	1278.89
Revenue of FB (RMB million)	258.60	255.88	8.25	1040.73
Other revenue (RMB million)	87.26	106.43	3.86	535.43

Note: All nominal variables are deflated by the GDP deflator with 2011 as the base year. FB, food and beverage industry.

**Table 2 ijerph-16-04319-t002:** Correlation Coefficients between Input and Output Variables.

	Revenue of Guest Rooms	Revenue of FB	Other Revenue
Number of employees	0.8691 (< 0.001)	0.9272 (< 0.001)	0.8047 (< 0.001)
Fixed assets	0.9628 (< 0.001)	0.8818 (< 0.001)	0.9417 (< 0.001)
Number of guest rooms	0.8734 (< 0.001)	0.88634 (< 0.001)	0.8257 (< 0.001)

Notes: Values in parentheses are *p* values. All correlation coefficients are significant at the 0.1% level.

**Table 3 ijerph-16-04319-t003:** Original DEA and Bias-corrected Efficiency Scores.

	Mean	Std.	Min	Max	DMUs
**Original DEA Efficiency Scores**					
Overall	0.7891	0.1637	0.4164	1.0000	124
East	0.8308	0.1732	0.4779	1.0000	52
Central	0.7604	0.1437	0.4164	1.0000	56
West	0.7541	0.1776	0.5098	1.0000	16
**Bias-Corrected Efficiency Scores**					
Overall	0.7208	0.1409	0.3906	0.9615	124
East	0.7688	0.1600	0.4496	0.9615	52
Central	0.6990	0.1148	0.3906	0.9474	56
West	0.6414	0.1050	0.4733	0.8185	16

Note: We used 200 replications for the first bootstrap of the double-bootstrapped procedure to obtain the bias-corrected efficiency scores. DEA, data envelopment analysis; DMUs, decision-making units.

**Table 4 ijerph-16-04319-t004:** Sample Means of Variables Used in the Bootstrapped Truncated Regression Model.

Variable	Definition	Mean	VIF
SO_2_	Sulfur dioxide (10,000 tons)	0.645	8.252
NO_x_	Nitrogen oxides (10,000 tons)	0.685	9.051
*S*&*D*	Smoke and dust (10,000 tons)	0.467	4.760
*ERev*	Entropy index of revenue diversification.	0.971	1.334
*RoadI*	Road intensity, distance of roads (km) divided by the area (10,000 km^2^) of provinces	9.151	3.465
*RailI*	Railroad intensity, distance of railroads (km) divided by the area (10,000 km^2^) of provinces	0.248	2.794
*DE*	1 if the province is in the east region; 0 otherwise	0.419	4.422
*DC*	1 if the province is in the central region; 0 otherwise	0.452	3.825
*Year*13	1 if the year is 2013; 0 otherwise	0.250	1.514
*Year*14	1 if the year is 2014; 0 otherwise	0.250	1.852
*Year*15	1 if the year is 2015; 0 otherwise	0.250	1.839

Note: All nominal variables are deflated by the GDP deflator with 2011 as the base year. VIF, variance inflation factor.

**Table 5 ijerph-16-04319-t005:** Truncated Regression Results.

Variables	Lower Bound			Mean	Upper Bound		
	0.5%	2.5%	5%		95%	97.5%	99.5%
Intercept	−0.4637	−0.1006	0.0367	0.8755 *	1.6717	1.8531	2.1957
SO_2_	−0.1873	−0.1003	−0.0471	0.2071	0.4560	0.5053	0.6006
NO_x_	−0.5549	−0.4793	−0.4235	−0.1686	0.0844	0.1429	0.2650
*S*&*D*	0.2810	0.3699	0.4120	0.625 ***	0.8327	0.8740	0.9380
*ERev*	−1.2147	−0.8538	−0.6824	0.1673	1.0509	1.2095	1.5923
*RoadI*	−0.0434	−0.0386	−0.0358	−0.0223 **	−0.0086	−0.0055	0.0013
*TrainI*	−1.3562	−1.2269	−1.1326	−0.6964 **	−0.1557	−0.0352	0.2806
*DE*	−0.1409	−0.0698	−0.0328	0.1227	0.2735	0.3003	0.3654
*DC*	−0.0383	0.0219	0.0461	0.1905 **	0.3264	0.3524	0.4199
*Year*13	−0.0140	0.0407	0.0618	0.1800 ***	0.2922	0.3122	0.3492
*Year*14	0.0768	0.1248	0.1497	0.2736 ***	0.3919	0.4188	0.4573
*Year*15	0.2000	0.2531	0.2794	0.4067 ***	0.5273	0.5444	0.5928
${\hat{\sigma }}_{\varepsilon }$	0.1768	0.1883	0.1944	0.2232 ***	0.2498	0.2543	0.2618

Notes: (1) *, **, and *** represent the 10%, 5%, and 1% levels of significance, respectively. (2) Mean (in column 5) is the average estimate obtained from maximum likelihood estimates and the bootstrap set with 2000 replications.

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
