# Peer review of "A Study of Different Types of Air Pollutants on the Efficiency of China’s Hotel Industry"

_ijerph, 2019, doi:10.3390/ijerph16224319_

Round 1
Reviewer 1 Report
General aspects
The topic is interesting and at first the use of DEA is appropriate.
Empirical research shows results showing that the impact of airborne pollutants is greater than that of invisible air pollutants on the operating performance of China's hotel industry. Uses DEA with bootstrapped truncated regression model.
Much needs to be improved regarding article submission, several non-standard Journal items.
-Improving the abstract needs to provide more information about the research and the method in brief. According to the paper template “We strongly encourage authors to use the following style of structured abstracts, but without headings: (1) Background: Place the question addressed in a broad context and highlight the purpose of the study; (2) Methods: Briefly describe the main methods or treatments applied; (3) Results: Summarize the article's main findings; and (4) Conclusions: Indicate the main conclusions or interpretations. ”
-Review the standards and formatting for the Journal.
-The sections of the article need to be numbered.
-Correct citations - You should quote only the number at the end, example | 1-3 |, do not need to put in parentheses.
Starts at line 33,
Line 61
The quotation on line 37 is correct.
Incorrect citations (non-journal standard) throughout the text.
-Reviewing 1st person use on line 67 “They”, impersonal is more appropriate for the scientific text.
-Line 162, source different from the others.
-The abbreviation F&B appears before (line 110) the authors present their meaning (line 198).
-In the methodology, the modeling steps are cited, however, using the same indicators of the bibliographic references, it needs to change
Methodology
The methodology was well presented, but only for the method and its generalities, but more details of the model created to solve the proposed problem should be presented.
For example, what are DMUs? not clearly seen in the article, should be presented in the methodology.
As for the results and discussion, I believe that the interpretation of the data could be better presented, increasing the perception between the model results and the authors conclusions.
Author Response
Authors’ Reply to Referee 1
We are grateful for your knowledgeable comments and are especially indebted to your kind patience and academic rigidity. Revisions and corrections have been made in accordance with your helpful and enlightening suggestions as follows.
Comment 1
Improving the abstract needs to provide more information about the research and the method in brief. According to the paper template “We strongly encourage authors to use the following style of structured abstracts, but without headings: (1) Background: Place the question addressed in a broad context and highlight the purpose of the study; (2) Methods: Briefly describe the main methods or treatments applied; (3) Results: Summarize the article's main findings; and (4) Conclusions: Indicate the main conclusions or interpretations.”
Reply 1
Thanks for your valuable suggestion. Based on your suggestions, we have rewritten the abstract (Please see line 12-21).
Comment 2
Review the standards and formatting for the Journal.
Reply 2
Thanks for reviewer’s valuable comments. Based on your suggestions, we have corrected the standards and formatted for the Journal.
Comment 3
The sections of the article need to be numbered.
Reply 3
Thanks for your valuable suggestion. We have numbered the sections.
Comment 4
Correct citations - You should quote only the number at the end, example | 1-3 |, do not need to put in parentheses. Starts at line 33, Line 61. The quotation on line 37 is correct. Incorrect citations (non-journal standard) throughout the text.
Reply 4
Thanks for reviewer’s valuable comments. Based on your suggestions, we have corrected citations throughout the text.
Comment 5
Reviewing 1st person use on line 67 “They”, impersonal is more appropriate for the scientific text.
Reply 5
Thanks for your valuable suggestion. We have changed “they” by “Simar and Wilson [26]” (Please see line 72-73).
Comment 6
Line 162, source different from the others.
Reply 6
Thanks for your valuable suggestion. We have corrected to the right form (Please see line 167).
Comment 7
The abbreviation F&B appears before (line 110) the authors present their meaning (line 198).
Reply 7
Thanks for reviewer’s valuable comments. We have defined food and beverage (F&B) at the first appearance (Please see line 107).
Comment 8
In the methodology, the modeling steps are cited, however, using the same indicators of the bibliographic references, it needs to change
Reply 8
Thanks for your valuable suggestion. Based on your suggestions, we used {} to represent the modeling steps.
Comment 9
The methodology was well presented, but only for the method and its generalities, but more details of the model created to solve the proposed problem should be presented.
For example, what are DMUs? not clearly seen in the article, should be presented in the methodology.
Reply 9
Thanks for reviewer’s valuable comments. Based on your suggestions, we have added one paragraph to describe why we use the bootstrapped truncated regression model (Please see line 119-129). In addition, we defined decision making units (DMUs) on line 130.
Comment 10
As for the results and discussion, I believe that the interpretation of the data could be better presented, increasing the perception between the model results and the authors conclusions.
Reply 10
Thanks for your valuable suggestion. Based on your suggestions, we have added more interpretations of the empirical results (Please see line 308-39, line 319-326)
Reviewer 2 Report
The introduction does not address persons who self-select out of visiting a destination because of severe allergies, respiratory illness and asthma. If you are aware that such pollution conditions exist at the destination, you may opt not to visit that location. This would suggest that persons visiting polluted destinations may have travel motives the may well overrule simple sightseeing.The paper refers to "efficiency" in China's hotel industry. I did not see a definition of efficiency and what is means for hotels in China. Does this include staff and the performance of their duties in the polluted environments? This would definitely affect efficiency?
I cannot speak to the methodology used in applying DEA, but levels of pollution may affect visitation which in turn might effect "efficiency". It might be meaningful to examine the areas of immediate impact as there might well be intervening variables not accounted for.
Reviewer 3 Report
The paper focuses on studying different types of air pollutants on the hotel and tourism industry in China. The topic is very interesting and appealing to the potential readers. The paper is based on the robust analysis and offers non-trivial results. It reads well and is written in plausible English.
However, there are some small issues I would like to point the authors’ attention at:
The Abstract is too short and concise. It does not offer the overview of the paper and is not informative. The Literature Review can be extended. More examples from China (if applicable) need to be mentioned. The dataset which was obtained from The Yearbook of China Tourism Statistics published by the National Tourism Administration of the People’s Republic of China during 2012-2015 and which consists of 31 provinces of China and 124 observations also raises some doubts. Is it representative? Are the results limited by the dataset? The Conclusions need to be extended (now, it is a repetition of the Abstract and Introduction). More insights, policy implications and own speculations are needed.
Author Response
Authors’ Reply to Referee 2
We are grateful for your knowledgeable comments and are especially indebted to your kind patience and academic rigidity. Revisions and corrections have been made in accordance with your helpful and enlightening suggestions as follows.
Comment 1
The Abstract is too short and concise. It does not offer the overview of the paper and is not informative.
Reply 1
Thanks for reviewer’s valuable comments. Based on your suggestions, we have rewritten the abstract (Please see line 12-21).
Comment 2
The Literature Review can be extended. More examples from China (if applicable) need to be mentioned.
Reply 2
Thanks for your valuable suggestion. Based on your suggestions, we have added more literatures, especially examples from China (Please see line 38-41)
Comment 3
The dataset which was obtained from The Yearbook of China Tourism Statistics published by the National Tourism Administration of the People’s Republic of China during 2012-2015 and which consists of 31 provinces of China and 124 observations also raises some doubts. Is it representative? Are the results limited by the dataset?
Reply 3
Thanks for reviewer’s valuable comments. The data set consists of 31 provinces. Since the study period includes 4 years (2012-2015), we have 124 observations (31 provinces * 4 years). In order to avoid confusion, we deleted statement “124 observations”.
Comment 4
The Conclusions need to be extended (now, it is a repetition of the Abstract and Introduction).
Reply 4
Thanks for reviewer’s valuable comments. Based on your suggestions, we have rewritten the conclusion (Please see line 328-335).
Comment 5
More insights, policy implications and own speculations are needed.
Reply 5
Thanks for reviewer’s valuable comments. Based on your suggestions, we have added more policy implications (Please see line 307-309, line 319-320).
Reviewer 4 Report
Dear Authors,
Thank you of the opportunity to review your manuscript. The topic of this paper is interesting and worthy of publication. In my opinion paper is well written and also provides interesting findings. However, I have a number of concerns with your work, and will outline those below
With all of these concerns, I believe you have major revisions ahead to make this submission appropriate for the journal as prestigious as the International Journal of Environmental Research and Public Health.
General:
Abstract:
The Abstract should be rewritten. There is lack of some necessary information. The abstract should be in a few sentences also give something of applications. Abstract is not just what is an article, but also shows the achievements (results). The abstract must contain the aim of the paper, a sentence about methods.
Furthermore, the first sentence of the Abstract is far too long. The reader can be easily lost while reading it. Also what do you mean by ‘a normal level air pollutants’? What is normal and acceptable in case of air pollution level in developing countries will not be unacceptable in many developed countries. Next sentence is also like a riddle to the reader. You wrote: ‘Hence, different types of air pollutants may have various impacts on local hotels’. I know what you mean, but it doesn’t flow from this sentence. After the first reading, I thought that your study will focus on some chemical impacts on hotels, its walls or something similar. You have to explain very clearly what you would like to do! In the last sentence of the Abstract, you should mention what was the most important achievement of your research. However, the reader can find very obvious finding. You wrote: ‘Our empirical research presents results showing that the impact of visible air pollutants is larger than that of invisible air pollutants on China’s hotel industry.’ Of course always the visible will have a bigger impact than invisible. And btw who/what you investigate? The visible air pollutant is larger than invisible, OK but who you investigate here - visitors’ perceptions, statistical data? Now, something is missing.
Introduction
You have conducted an interesting study and we need to make sure that readers will puzzle it.
Your Introduction is a bit messy. Please note that proper Introduction should have half (max. one) page (the rest of the information you can add as a Background), and have a lack of proper structure. Literally speaking, the Introduction must answer the questions: What was I studying? Why was it an important question? What did we know about it before I did this study? And, How will this study advance our knowledge? Try to focus on a wider perspective and use your case study as an example.
The first part (about tourism number, jobs in tourism sector) is completely not necessary. The same suggestion is to last part of your introduction, i.e. ‘The rest of the paper is organized as follows. Section II introduces hypotheses and the research framework. Section III describes the bootstrapped truncated regression model. Section IV consists of the description of the data and the variables, the empirical results, and managerial implications. The final section offers concluding remarks’. This should be deleted.
In some sentences, like e.g., ‘China is noted for its beautiful scenery, abundant heritage resources, and distinctive cultures and since 1990 has been the third largest inbound tourism destination, even becoming the second largest tourism market in 2018’ a citation is needed.
Overall, please follow those questions I’ve mentioned and your Introduction will be more logical and much more easier in understanding.
Background
I would like to recommend you to develop a proper background. You can put here what you mention at Introduction and Hypotheses and Research Framework
This part should begin with defining a topic to a wider audience. Thus, the background of your study will provide context to the information discussed throughout the research paper.
Furthermore, this section should discuss the theoretical aspects by involving the background of the theories published previously in the research literature and also focus on the ambiguities arose in these works.
Methodology
The methodology is well written, and reader can fallow your logic from first to last sentence of this part. Well done.
Empirical Analysis and Discussion
This part you should be rewritten. You are mixing the Results (at your paper Empirical Analysis) with the Discussion, at some parts of your Results you conduct Discussion, and this is improper. At this part, i.e., Empirical Analysis you should only present your findings.
Discussion part is missing in this paper. That is why I recommend you follow my suggestions to develop proper Introduction and Background. This will allow you to conduct proper Discussion. The purpose of the discussion is to interpret and describe the significance of your findings in light of what was already known about the research problem being investigated, and to explain any new understanding or fresh insights about the problem after you've taken the findings into consideration. The proper discussion should be connected to the introduction. Thus proper Introduction and Background is necessary.
Other issues (Line number of the comments refer to the line numbers added by the Editorial System):
Data envelopment analysis (DEA)… - Data Envelopment Analysis (DEA)… (Kneip et al. [43]) – should be just [43] (Baker [44]) – should be just [44] Simar and Wilson (2007) – should be – Simar and Wilson [20] The dataset, obtained from The Yearbook of China Tourism Statistics [citation needed] SO2 – should be SO2 same Table 5. Environment Canada [53] claimed that tourism is one of the two industries that can be damaged when air quality is poor.>>> what is the second?
Looking forward to see you revised manuscript.
Author Response
Authors’ Reply to Referee 3
We are grateful for your knowledgeable comments and are especially indebted to your kind patience and academic rigidity. Revisions and corrections have been made in accordance with your helpful and enlightening suggestions as follows.
Comment 1
The Abstract should be rewritten. There is lack of some necessary information. The abstract should be in a few sentences also give something of applications. Abstract is not just what is an article, but also shows the achievements (results). The abstract must contain the aim of the paper, a sentence about methods.
Furthermore, the first sentence of the Abstract is far too long. The reader can be easily lost while reading it. Also what do you mean by ‘a normal level air pollutants’? What is normal and acceptable in case of air pollution level in developing countries will not be unacceptable in many developed countries. Next sentence is also like a riddle to the reader. You wrote: ‘Hence, different types of air pollutants may have various impacts on local hotels’. I know what you mean, but it doesn’t flow from this sentence. After the first reading, I thought that your study will focus on some chemical impacts on hotels, its walls or something similar. You have to explain very clearly what you would like to do! In the last sentence of the Abstract, you should mention what was the most important achievement of your research. However, the reader can find very obvious finding. You wrote: ‘Our empirical research presents results showing that the impact of visible air pollutants is larger than that of invisible air pollutants on China’s hotel industry.’ Of course always the visible will have a bigger impact than invisible. And btw who/what you investigate? The visible air pollutant is larger than invisible, OK but who you investigate here - visitors’ perceptions, statistical data? Now, something is missing.
Reply 1
Thanks for reviewer’s valuable comments. Based on your suggestions, we have rewritten the abstract (Please see line 12-21).
Comment 2
The first part (about tourism number, jobs in tourism sector) is completely not necessary. The same suggestion is to last part of your introduction, i.e. ‘The rest of the paper is organized as follows. Section II introduces hypotheses and the research framework. Section III describes the bootstrapped truncated regression model. Section IV consists of the description of the data and the variables, the empirical results, and managerial implications. The final section offers concluding remarks’. This should be deleted.
Reply 2
Thanks for reviewer’s valuable comments. Based on your suggestions, we have deleted these two paragraphs.
Comment 3
Your Introduction is a bit messy. Please note that proper Introduction should have half (max. one) page (the rest of the information you can add as a Background), and have a lack of proper structure. Literally speaking, the Introduction must answer the questions: What was I studying? Why was it an important question? What did we know about it before I did this study? And, How will this study advance our knowledge? Try to focus on a wider perspective and use your case study as an example.
Reply 3
Thanks for reviewer’s valuable comments. Based on your suggestions, we have rewritten the first paragraph in the introduction section (Please see line 26-35).
Comment 4
In some sentences, like e.g., ‘China is noted for its beautiful scenery, abundant heritage resources, and distinctive cultures and since 1990 has been the third largest inbound tourism destination, even becoming the second largest tourism market in 2018’ a citation is needed.
Reply 4
Thanks for reviewer’s valuable comments. Based on your suggestions, we have added citations (Please see line 37-38).
Comment 5
I would like to recommend you to develop a proper background. You can put here what you mention at Introduction and Hypotheses and Research Framework. This part should begin with defining a topic to a wider audience. Thus, the background of your study will provide context to the information discussed throughout the research paper. Furthermore, this section should discuss the theoretical aspects by involving the background of the theories published previously in the research literature and also focus on the ambiguities arose in these works.
Reply 5
Thanks for reviewer’s valuable comments. Based on your suggestions, we have added some background in the introduction section (Please see line 26-35, line 38-41, line 49-59).
Comment 6
The methodology is well written, and reader can fallow your logic from first to last sentence of this part. Well done.
Reply 6
Thank you.
Comment 7
This part you should be rewritten. You are mixing the Results (at your paper Empirical Analysis) with the Discussion, at some parts of your Results you conduct Discussion, and this is improper. At this part, i.e., Empirical Analysis you should only present your findings.
Discussion part is missing in this paper. That is why I recommend you follow my suggestions to develop proper Introduction and Background. This will allow you to conduct proper Discussion. The purpose of the discussion is to interpret and describe the significance of your findings in light of what was already known about the research problem being investigated, and to explain any new understanding or fresh insights about the problem after you've taken the findings into consideration. The proper discussion should be connected to the introduction. Thus proper Introduction and Background is necessary.
Reply 7
Thanks for reviewer’s valuable comments. Based on your suggestions, we have added the discussion section and connected the introduction (Please see line 298-326).
Comment 8
Data envelopment analysis (DEA)… - Data Envelopment Analysis (DEA)… (Kneip et al. [43]) – should be just [43] (Baker [44]) – should be just [44] Simar and Wilson (2007) – should be – Simar and Wilson [20]
Reply 8
Thanks for reviewer’s valuable comments. Based on your suggestions, we have changed to the right citations.
Comment 9
The dataset, obtained from The Yearbook of China Tourism Statistics [citation needed]
Reply 9
Thanks for reviewer’s valuable comments. Based on your suggestions, we have added the citations (Please see line 200).
Comment 10
SO2 – should be SO2 same Table 5.
Reply 10
Thanks for reviewer’s valuable comments. Based on your suggestions, we have changed the right form (Please see line 239, line 287, Table 4, Table 5).
Comment 11
Environment Canada [53] claimed that tourism is one of the two industries that can be damaged when air quality is poor.
>>> what is the second?
Reply 11
The web does not exist now and thus we cannot answer this question. Hence, we have deleted this sentence and the corresponding citation.
Round 2
Reviewer 1 Report
Thanks for making the suggested corrections, the text has a much better presentation.
Just one observation equation indicator (1) appears for two different equations on lines 133 and 148.
Author Response
Comment 1
Thanks for making the suggested corrections, the text has a much better presentation.
Just one observation equation indicator (1) appears for two different equations on lines 133 and 148.
Reply 1
Thanks for your valuable suggestion. We have corrected the equation number (Please see line 131 and 146).
Reviewer 4 Report
Dear Author(s),
Thank you of the opportunity to review your resubmitted manuscript. Like I’ve mentioned previously the topic of this paper is interesting and worthy of publication. However, still I have a some concerns with your work, and will outline those below.
With these concerns, I believe you have major/minor revisions ahead to make this submission appropriate for the journal as prestigious as the International Journal of Environmental Research and Public Health.
Abstract:
It is much better than it was but still, first and second sentence together have no sense at all.
Please check the version below:
The literature generally agrees about the negative impacts of air pollution on tourism. This study focuses on visible and invisible air pollutants and their impacts on China’s hotel industry. Overall, the visible air pollutants may block the sights and sceneries and worsen the quality of visitors’ sensory sensations, and invisible air pollutants are unlikely to result in the same perceptions and sensations. Hence, different types of air pollutants may have various impacts on the operational performance of the hotel industry. This study employed the bootstrapped truncated regression model to investigate whether different types of air pollutants have distinctive impacts on the hotel industry. The dataset consists of 31 provinces of China for the period 2012-2015. Empirical results indicated that visible air pollutants significantly decrease the operational efficiency of China’s hotel industry, while invisible air pollutants insignificantly affect the hotel industry.
Introduction
Introduction is fine now, however there is lack of literature. Every sentence, like this below, must have (!) quotation.
Air quality can influence the process of making travel decisions as well as shape the competitiveness of tourist destinations [?].
Discussion
5. Discussion(s) – erase ‘s’ !Discussion!
I can see that you improve this section a little bit, however, it is not enough in my opinion. You conducted very interesting research supported by a proper methodology, however, due to short discussion all good things are lost somewhere.
Once again: The purpose of the discussion is to interpret and describe the significance of your findings in light of what was already known about the research problem being investigated, and to explain any new understanding or fresh insights about the problem after you've taken the findings into consideration.
Furthermore, at the end, you should at least mention, what kind recommendation you propose.
327 6. Conclusions – erase !s! Conclusion
The Future Research section would also be nice.
Looking forward to see you revised manuscript.
Author Response
Comment 1
It is much better than it was but still, first and second sentence together have no sense at all. Please check the version below:
The literature generally agrees about the negative impacts of air pollution on tourism. This study focuses on visible and invisible air pollutants and their impacts on China’s hotel industry. Overall, the visible air pollutants may block the sights and sceneries and worsen the quality of visitors’ sensory sensations, and invisible air pollutants are unlikely to result in the same perceptions and sensations. Hence, different types of air pollutants may have various impacts on the operational performance of the hotel industry. This study employed the bootstrapped truncated regression model to investigate whether different types of air pollutants have distinctive impacts on the hotel industry. The dataset consists of 31 provinces of China for the period 2012-2015. Empirical results indicated that visible air pollutants significantly decrease the operational efficiency of China’s hotel industry, while invisible air pollutants insignificantly affect the hotel industry.
Reply 1
Thanks for your valuable suggestion. Based on your suggestions, we have rewritten the abstract (Please see line 12-20).
Comment 2
Introduction is fine now, however there is lack of literature. Every sentence, like this below, must have (!) quotation.
Air quality can influence the process of making travel decisions as well as shape the competitiveness of tourist destinations [?].
Reply 2
Thanks for reviewer’s valuable comments. We have added appropriate citations (Please see line 26-28).
Comment 3
Discussion(s) – erase ‘s’ !Discussion!Reply 3
Based on your suggestion, we have changed from “5. Discussions” to “5. Discussion” (Please see line 294)
Comment 4
I can see that you improve this section a little bit, however, it is not enough in my opinion. You conducted very interesting research supported by a proper methodology, however, due to short discussion all good things are lost somewhere.
Once again: The purpose of the discussion is to interpret and describe the significance of your findings in light of what was already known about the research problem being investigated, and to explain any new understanding or fresh insights about the problem after you've taken the findings into consideration.
Furthermore, at the end, you should at least mention, what kind recommendation you propose.
Reply 4
Thanks for reviewer’s valuable comments. Based on your suggestions, we have added more policy analysis in the Discussion (Please see line 294-338).
Comment 5
Conclusions – erase !s! ConclusionReply 5
Based on your suggestion, we have changed from “6. Conclusions” to “6. Conclusion” (Please see line 339)
Comment 5
The Future Research section would also be nice.
Reply 5
Thanks for reviewer’s valuable comments. Based on your suggestions, we have added the future research in the conclusion (see line 348-353).